# Embedding and Backscattered Scanning Electron Microscopy (EM-BSEM) Is Preferential over Immunophenotyping in Relation to Bioprosthetic Heart Valves

**DOI:** 10.3390/ijms241713602

**Published:** 2023-09-02

**Authors:** Alexander Kostyunin, Tatiana Glushkova, Elena Velikanova, Rinat Mukhamadiyarov, Leo Bogdanov, Tatiana Akentyeva, Evgeny Ovcharenko, Alexey Evtushenko, Daria Shishkova, Yulia Markova, Anton Kutikhin

**Affiliations:** Department of Experimental Medicine, Research Institute for Complex Issues of Cardiovascular Diseases, 6 Sosnovy Boulevard, Kemerovo 650002, Russia; kostae@kemcardio.ru (A.K.); glushtv@kemcardio.ru (T.G.); veliea@kemcardio.ru (E.V.); muhara@kemcardio.ru (R.M.); bogdla@kemcardio.ru (L.B.); akentn@kemcardio.ru (T.A.); ovchea@kemcardio.ru (E.O.); ave@kemcardio.ru (A.E.); shidk@kemcardio.ru (D.S.); markyo@kemcardio.ru (Y.M.)

**Keywords:** bioprosthetic heart valves, structural valve degeneration, calcific aortic stenosis, calcific aortic valve disease, cell plasticity, immunophenotyping, endothelial-to-mesenchymal transition, extracellular matrix degradation, valvular interstitial cells, macrophages

## Abstract

Hitherto, calcified aortic valves (AVs) and failing bioprosthetic heart valves (BHVs) have been investigated by similar approaches, mostly limited to various immunostaining techniques. Having employed multiple immunostaining combinations, we demonstrated that AVs retain a well-defined cellular hierarchy even at severe stenosis, whilst BHVs were notable for the stochastic degradation of the extracellular matrix (ECM) and aggressive infiltration by ECM-digesting macrophages. Leukocytes (CD45^+^) comprised ≤10% cells in the AVs but were the predominant cell lineage in BHVs (≥80% cells). Albeit cells with uncertain immunophenotype were rarely encountered in the AVs (≤5% cells), they were commonly found in BHVs (≥80% cells). Whilst cell conversions in the AVs were limited to the endothelial-to-mesenchymal transition (represented by CD31^+^α-SMA^+^ cells) and the formation of endothelial-like (CD31^+^CD68^+^) cells at the AV surface, BHVs harboured numerous macrophages with a transitional phenotype, mostly CD45^+^CD31^+^, CD45^+^α-SMA^+^, and CD68^+^α-SMA^+^. In contrast to immunostaining, which was unable to predict cell function in the BHVs, our whole-specimen, nondestructive electron microscopy approach (EM-BSEM) was able to distinguish between quiescent and matrix-degrading macrophages, foam cells, and multinucleated giant cells to conduct the ultrastructural analysis of organelles and the ECM, and to preserve tissue integrity. Hence, we suggest EM-BSEM as a technique of choice for studying the cellular landscape of BHVs.

## 1. Introduction

Despite significant efforts made to develop therapeutics to prevent or treat calcific aortic stenosis (CAS), a disease characterised by pathological fibrotic thickening and the calcification of aortic valve (AV) cusps, which then become incapable of properly regulating cardiac outflow, the replacement of the AVs with mechanical (MHVs) or bioprosthetic heart valves (BHVs) remains the only evidence-based approach to restore valvular function [1]. Annually, from 280,000 to 370,000 AV replacements are carried out worldwide [2,3], and BHVs are preferrable over MHVs in elderly patients because of their excellent haemocompatibility and better blood flow pattern [4]. However, BHVs are prone to structural valve degeneration (SVD), which limits their lifespan to 15–20 years, requires a redo surgery in 20% to 50% of the patients at 15 years postimplantation, and restricts the choice of the patients to those ≥60 years of age, since their average expected lifespan does not exceed that of the BHV [1,5,6]. SVD is triggered by the enzymatic degradation of the xenopericardium, as, opposite to the AVs, decellularised BHVs lack proper extracellular matrix (ECM) repair and cannot perform an adequate reconstruction of ECM fibres in response to the enzymes precipitating from circulation or released by infiltrating host immune cells [7,8]. Binding of Ca^2+^ ions to damaged ECM fibres induces mineralisation, which is also uncontrolled and promotes the deterioration of the prosthetic ECM [7,8]. Therefore, SVD represents a progressive and irreversible degeneration of the BHV, eventually leading to its destruction [5,6,7,8]. Delamination, fragmentation, and calcification of collagen fibres observed at the initial stages of SVD result in the disintegration of the ECM and evolve into perforations, tears, and mineral deposits within the leaflets, eventually leading to transprosthetic regurgitation and repeated stenosis [5,6,7,8]. To summarise, enzymatic degradation and calcification result in the inability of BHVs to regain function of the native AV, and BHV failure recapitulates the clinical scenario observed in CAS [7,8].

Among the hallmarks of SVD is combined macrophage and neutrophil infiltration, where macrophages digest the xenogeneic ECM and neutrophils eliminate bacteria, which invade the BHVs during the transient bacteriemia [9,10,11,12,13]. Other cell populations within the BHVs include endothelial cells (CD31^+^) and myofibroblasts (α-SMA^+^) [14,15,16,17], though their appearance and localisation differ from AVs, which are composed of valvular endothelial cells (VECs) and valvular interstitial cells (VICs) [18,19]. Whilst even stenotic AVs, which contain excessive amounts of the ECM and ectopic calcium deposits, still retain their cytoarchitecture [18,19], the distribution of endothelial cells, myofibroblasts, and immune cells in failing BHVs is uneven, and macrophages within the degraded xenogeneic ECM show significant heterogeneity [8]. Albeit calcified AVs and failing BHVs have striking differences in their cellular and molecular composition, the algorithms for their histological analysis have been largely similar to date, obscuring the causes of SVD.

To better understand the molecular differences in valvular cell populations in CAS and SVD, we performed an immunophenotyping and electron microscopy examination of calcified AVs and failing BHVs. In excised stenotic AVs, VECs were reliably detected by canonical markers (CD31 and VEGFR2), activated VICs were notable for the high expression of alpha-smooth muscle actin (α-SMA) and vimentin (VIM), and infiltrating leukocytes were distinguishable by CD45^+^ staining. In contrast, the majority of cells within the BHVs had an intermediate phenotype, combining markers of ECs and myofibroblasts (CD31 and α-SMA), ECs and macrophages (CD31 and CD68), or myofibroblasts and leukocytes (α-SMA, CD45, and CD68). Whereas immunophenotyping was unable to address the pathophysiological significance of BHV-infiltrating macrophages, our modified EM-BSEM (EMbedded and Backscattered Scanning Electron Microscopy) approach (which is based on consecutive heavy-metal staining, epoxy-resin embedding, the grinding and polishing of epoxy resin blocks, and backscattered scanning electron microscopy) detected the matrix-degrading activity of macrophages, indicative of their functional specification and pathogenic effects. Further, EM-BSEM clearly visualised other host-cell populations within the BHVs (i.e., neutrophils, myofibroblasts, and endothelial-like cells), as well as degraded and loosened ECM fibres, permitted the ultrastructural analysis of cytoplasmic content, and retained calcium deposits in the excised valves without the loss of their integrity. Having demonstrated the transitional phenotype of most immune cells within the BHVs for the first time, we suggest EM-BSEM as an appropriate and superior technique to study the causes of BHV failure.

## 2. Results

### 2.1. Histological, Immunohistochemical, and Electron Microscopy Examination of Calcified AVs and Failing BHVs Shows the Advantages of Ultrastructural Analysis by the EM-BSEM Approach

We first interrogated excised AVs (*n* = 4) and BHVs (*n* = 4) employing haematoxylin and eosin (H&E) staining to reveal the causes of their failure. AVs displayed severe fibrosis and contained multiple mineral deposits, which were primarily located at the base of the leaflets near the aortic side (Figure 1A). Nevertheless, ECM fragmentation in AVs was limited to superficial layers and the immune cells were mainly associated with calcifications and a degraded ECM (Figure 1A). Leaflets of the BHVs were also thickened and calcified, but the ECM degradation involved the entire leaflet and resulted in delaminations and tears (Figure 1B). Host cells were sparsely distributed across the BHVs, and were focused at or beneath the leaflet surface and around the mineral deposits and perforations (Figure 1B).

Next, we investigated cell populations and the ECM of the AVs and BHVs using EM-BSEM, a recently developed ultrastructural pathology technique for the examination of calcified tissues which relies on consecutive heavy-metal staining, epoxy-resin embedding, grinding, polishing, and backscattered scanning electron microscopy to visualise entire tissue segments, and by means of conventional immunohistochemical staining. By immunohistochemistry, we examined the distribution of CD31 (type I glycosylated transmembrane protein, a marker of endothelial cells), α-SMA (a contractile protein and a marker of myofibroblasts and smooth muscle cells), CD45 (type I glycosylated transmembrane protein, a panleukocyte marker), and CD68 (another type I glycosylated transmembrane protein, a marker of macrophages). The surface of the AVs was covered by a monolayer of typical, elongated, CD31^+^ VECs with a large centred oval nucleus, whilst the BHVs were lined with irregular endothelial-like cells which had a lower nucleus-to-cytoplasm ratio, multiple electron-dense inclusions within the cytoplasm, and were slightly positive for CD31 (Figure 2A). The phenotype of the VECs was perfectly reported by both imaging modalities, yet EM-BSEM was significantly more informative when applied to the BHVs by highlighting the cytoplasmic content (Figure 2A). Areas of the intact and fibrotic ECM alternated within the thickened AVs and contained numerous α-SMA^+^ VICs with oblong nuclei and a high nucleus-to-cytoplasm ratio (Figure 2B). In contrast, myofibroblasts were sparse in the loosened ECM of the BHVs, generally displaying lower α-SMA expression (Figure 2B). Regardless of the valve type, CD45^+^ and CD68^+^ macrophages were restricted to calcium deposits and a fibrotic (AVs) or degraded (BHVs) ECM (Figure 2C). As opposed to the canonical macrophages within the AVs, macrophages within the BHVs had large round or pear-shaped nuclei and multiple different granules in the cytoplasm (attesting them as activated ECM-digesting macrophages) or several nuclei and similar granular patterns categorising them as multinucleated foreign-body giant cells (Figure 2C). In comparison with immunohistochemical staining, the ultrastructural EM-BSEM approach was capable of assessing the ECM-digesting activity and clearly visualised the ECM degradation and fragmentation (Figure 2C). All studied markers (CD31, α-SMA, CD45, and CD68) were associated with the host cells within the degraded ECM of the BHVs, although endothelial-like cells and myofibroblasts were infrequent in the electron microscopy visualisation (Figure 2C). In addition to macrophages, BHVs were invaded by neutrophils, in agreement with our previous findings [18].

Hence, immune-cell markers within the AVs corresponded to the electron microscopy data, and the cellular composition of the calcified AVs accorded to the pattern reported in the literature [20], confirming the high relevance of immunostaining and EM-BSEM in relation to the AVs. However, immunophenotyping of the BHVs was unable to clearly identify cell populations, as CD31^+^, α-SMA^+^, CD45^+^, and CD68^+^ cells were frequently detected over the entire BHV section and were generally associated the degraded ECM (Figure 2A–C). The promiscuous distribution of immune-cell markers within the BHVs was not concordant with the electron microscopy analysis results, where endothelial-like cells were found at the BHV surface and numerous macrophages prevailed over rare myofibroblasts.

To further explore the opportunities of the ultrastructural analysis in relation to the BHVs, we analysed the host-cell populations and the degraded ECM at a high magnification. Whereas the VECs retained cell–cell contacts, thereby forming a continuous monolayer even in calcified and failing AVs, endothelial-like cells in degenerated BHVs were frequently found in close association with adhering monocytes, and such a layer was discontinuous (Figure 3A). The VICs within the AVs had a high nucleus-to-cytoplasm ratio and did not contain inclusions, in contrast to the mesenchymal cells (represented by fibroblasts and myofibroblasts) in the BHVs, which engulfed and aggregated lipid droplets (Figure 3B). An evident advantage of EM-BSEM was an ability to assess the ECM-digesting activity of invading macrophages within both the AVs and BHVs, although the number of macrophages and the amounts of degraded ECM in their vicinity was considerably higher within the latter (Figure 3C).

### 2.2. Immunostaining Well Defines the Phenotype of Cell Populations within the AVs but Fails to Tackle This Task in BHVs

To better understand the distribution of endothelial (CD31 and VEGFR2), contractile (α-SMA and VIM), and immune-cell (CD45 and CD68) markers within the AVs and BHVs, we employed immunofluorescence staining, which generally permits the simultaneous, sensitive, and specific detection of several molecules. In keeping with immunohistochemistry, CD31^+^/VEGFR2^+^ cells (annotated as VECs) were observed at the surface of the AVs (Figure 4A), whereas α-SMA^+^ and VIM^+^ cells (defined as VICs) were common within the leaflets (Figure 4B), and areas around calcium deposits were densely populated with CD45^+^CD68^+^ or CD45^+^CD68^-^ immune cells (Figure 4C), in keeping with the seminal papers [21]. On the contrary, BHVs contained large foci of CD31^+^VEGFR2^-^ (but not CD31^+^VEGFR2^+^ cells; Figure 4D), α-SMA^+^VIM^-^ (but not α-SMA^+^VIM^+^ cells; Figure 4E), and CD45^+^CD68^+^ cells (Figure 4F) across the valve in association with the degraded ECM.

Since CD31^+^, α-SMA^+^, CD45^+^, and CD68^+^ cells had similar localisation patterns in the ECM of the BHVs, we stained both types of the valves for different cell-state transitions. Staining of the AVs found a substantial proportion of VECs which coexpressed CD31 and α-SMA (Figure 5A), suggestive of the endothelial-to-mesenchymal transition (EndoMT), a state which is frequently observed in patients with CAS [22,23]. CD31^+^α-SMA^+^ cells lined the AV surface and did not migrate deeper into the AV (Figure 5A). Albeit AVs did not show any CD31^+^CD45^+^ cells (Figure 5B), there have been single CD31^+^CD68^+^ cells with large nuclei at the AV surface (Figure 5C), which probably corresponded to endothelial-like cells or monocytes that were previously observed at the surface of the BHVs. Earlier, CD31 expression in circulating monocytes were reported in patients with cancer [24,25]. Notably, CD68^+^ cells located in the subendothelial layer and around calcium deposits did not express CD31 (Figure 5C). Further, Avs did not contain α-SMA^+^CD45^+^ cells (Figure 5D), but sparse α-SMA^+^CD68^+^ cells reminiscent of VICs which acquired macrophage markers (Figure 5E) were identified. CD68^+^ cells around calcium deposits and the degraded ECM did not display α-SMA (Figure 5E).

Although double-positive cells were rare (≤5%) in the AVs, they comprised ≥80% cells within the BHVs. As such, 68.2% cells in the BHVs were positive for CD31 and α-SMA (CD31^+^α-SMA^+^, Figure 6A and Figure 7A), 79.1% cells were positive for CD31 and CD45 (CD31^+^CD45^+^, Figure 6B and Figure 7B), 13.1% cells were positive for CD31 and CD68 (CD31^+^CD68^+^, Figure 6C and Figure 7C), 23.5% cells were positive for α-SMA and CD45 (α-SMA^+^CD45^+^, Figure 6D and Figure 7D), and 73.9% cells were positive for CD68 and α-SMA (CD68^+^α-SMA^+^, Figure 6E and Figure 7E). The majority of double-positive cells infiltrated the degraded ECM of the BHVs (Figure 6A–E).

### 2.3. Quantitative Image Analysis Demonstrates That the Majority of Host Cells in the BHVs Have Mixed Immunophenotype

Comparison of the frequencies of cell populations within each staining combination in relation to the total count of positively stained cells (within the respective images) revealed that α-SMA^+^CD31^−^, α-SMA^+^CD45^−^, and α-SMA^+^CD68^−^ prevailed in the AVs (82.9%, 91.1%, and 94.3%, respectively), but were occasional in the BHVs (3.3%, 10.3%, and 5.8%, respectively, Figure 7A,D,E). Contrariwise, CD68^+^CD31^−^, CD45^+^α-SMA^−^, and CD68^+^α-SMA^−^ cells were frequent in the BHVs (68.1%, 50.4%, and 18.3%, respectively), but absent or sparse in the AVs (0.0%, 8.9%, and 2.5%, respectively, Figure 7C–E). Hence, VICs (α-SMA^+^CD31^−^, α-SMA^+^CD45^−^, and α-SMA^+^CD68^−^ cells) prevailed over VECs and macrophages in the AVs, and macrophages (CD68^+^CD31^−^, CD45^+^α-SMA^−^, and CD68^+^α-SMA^−^ cells) were the principal cell population in the BHVs. The relative proportion of CD31^+^α-SMA^−^ and CD31^+^CD45^−^ VECs in the AVs were similar (15.0% and 17.7%, respectively, Figure 7A,B). In relation to CD31/CD68 staining, CD31^+^CD68^−^ cells (VECs) prevailed in the AVs (76.6%), whilst CD68^+^CD31^−^ cells (macrophages) were predominant in the BHVs (68.1%, Figure 7C).

For better a characterisation of the cell composition in the AVs and BHVs, we evaluated the proportions of CD31^+^/CD45^+^, α-SMA^+^/CD45^+^, CD31^+^/CD68^+^, and α-SMA^+^/CD68^+^ cells in relation to the total count of CD45^+^ and CD68^+^ cells, respectively. In the BHVs, almost all (99.1%) CD45^+^ cells expressed CD31 and 30.2% of CD45^+^ cells were also α-SMA^+^, but these phenomena have not been observed in the AVs (Figure 8A). The proportions of CD68^+^/CD31^+^ cells did not differ significantly between the AVs and BHVs (18.4% and 14.1% of all CD68^+^ cells, respectively, Figure 8B), but 79.2% of all CD68^+^ cells in the BHVs were also α-SMA^+^, whereas none of the CD68^+^/α-SMA^+^ cells were detected in the AVs (Figure 8B).

## 3. Discussion

Although both CAS and SVD affect the heart valves in the same anatomical location, are accompanied by valvular calcification, and eventually require heart-valve replacement (primary and redo surgery, respectively), these diseases have a largely distinct pathophysiology [18]. CAS, which affects AVs, is induced by endothelial dysfunction, lipid retention, and macrophage infiltration, which collectively result in the inflammation and increased deposition of the fibrotic ECM [18,19]. At the late stage of CAS, an excessive ECM becomes unevenly distributed across the AV, thus deteriorating its biomechanical properties [18,19]. A disturbed microenvironment provokes the osteogenic reprogramming of VICs, which produce a procalcific milieu that ultimately leads to the formation of large calcium deposits, which make the AVs incapable of coping with high-velocity blood flow which is expelled from the left ventricle to the aorta [18,19]. Taken together, the pathological thickening and calcification of the AV results in aortic stenosis or regurgitation, which determine the clinical symptoms of CAS [18,19].

However, SVD is caused by the enzymatic degradation of the bovine or porcine pericardium, which is commonly used for the fabrication of BHVs [8,18]. Circulating proteases cause the loosening of the ECM, which promotes the adhesion of neutrophils and monocytes to the valvular surface and their migration into the deeper layers of xenopericardium [8,18]. Cell-derived proteases further contribute to the digestion of ECM components that enhances the deposition of Ca^2+^ ions onto the damaged collagen fibres [8,18]. As currently employed BHVs are unable to inhibit or block circulating proteases and Ca^2+^ ions, as well as invading immune cells, being also devoid of any repair mechanisms, enzymatic degradation and calcification lead to ECM disintegration, perforations and tears within the xenopericardium, SVD, and BHV failure [8]. 

Yet, our understanding of SVD mechanisms remains incomplete and requires implementation of ultrastructural and high-throughput approaches. It is widely accepted that SVD is inevitable in all existing BHV models, since devitalisation of bovine or porcine tissues abrogates any inherent regeneration, whilst enzymatic degradation and calcification develop over time [8]. The lifespan of BHVs primarily depends on crosslinks introduced by glutaraldehyde or the ethylene glycol diglycidyl ether fixation of xenogeneic tissue, though it is also negatively impacted by young age, prosthesis–patient mismatch, arterial hypertension, dyslipidemia, diabetes mellitus, metabolic syndrome, chronic kidney disease, increased levels of calcium-phosphate product, lipoprotein-associated phospholipase A2 and proprotein convertase subtilisin/kexin type 9 in plasma, and the absence of hypolipidemic and anticoagulant therapy [7,26,27,28,29,30,31,32,33,34,35]. A diversity of SVD risk factors implies a variety of mechanisms behind this disease, among which are: (1) material fatigue upon cyclic load; (2) the precipitation of circulating proteases (e.g., matrix metalloproteinases and cathepsins) digesting ECM components; (3) the retention of oxidated low-density lipoprotein cholesterol provoking oxidative stress; (4) the deposition of advanced glycation end-products promoting immune-cell infiltration; (5) immune-cell attack caused by galactose-α1,3-galactose and N-glycolylneuraminic acid, which bear immunogenic xenoglycans; (6) dystrophic calcification promoted by the abundance of nucleation sites (e.g., damaged collagen fibres) and deficiency in acidic serum proteins serving as mineral chaperones [8,11,13,18,35,36,37]. However, the evolution of BHVs is considerably hindered by the cost of growing genetically modified animals free of immunogenic xenoglycans and the susceptibility of alternative chemical fixation regimens to the abovementioned factors.

Despite the mechanisms of CAS and SVD being strikingly different, these diseases are commonly studied by similar molecular and cellular-imaging modalities, such as immunohistochemical or immunofluorescence staining, which furnish an opportunity to distinguish between cellular phenotypes through the established molecular signatures and to pinpoint key cell populations. Here, we revealed that, whilst being highly relevant for the immunophenotyping of the AVs, this approach is less informative for investigating BHVs, as most of the cell populations there are positive for several markers, which, in this case, become nonspecific. Whereas VICs (CD31^−^CD45^−^αSMA^+^) and VECs (CD31^+^CD45^−^αSMA^−^) are two primary cell populations in the AVs (≈90% cells if counted together), leukocytes (CD45^+^) comprise ≤ 10% cells in the AVs but constitute ≥80% cells in the BHVs. Such differences are mostly related to the origin of the AVs (tissue evolved through the ontogenesis with a defined cellular hierarchy, which becomes altered with age) [20] and BHVs (xenogeneic decellularised tissue which is infiltrated by circulating immune cells upon the implantation) [8]. Moreover, the neutrophil or macrophage attacks of the BHVs are promoted by transient bacteriemia (i.e., the migration of opportunistic pathogens into the blood) or residual xenoantigens (e.g., galactose-alpha-1,3-galactose or N-glycolylneuraminic acid), respectively, collectively resulting in the inhabitation of the BHVs by immune cells [38,39]. In keeping with these findings, transitional phenotype was an infrequent phenomenon in the AVs (≤5% cells positively stained for markers of distinct cell lineages) but was common for the BHVs (≥80% cells), where host cells dwell in an atypical microenvironment composed of a crosslinked xenogeneic ECM and have an unlimited access to various biochemical cues from the circulating blood. In combination, these conditions may induce cell reprogramming and a wide array of growth factors and cytokines might increase cell plasticity, thus promoting the development of several transitional phenotypes. Digestion of ECM fibres by circulating and macrophage-derived enzymes leads to the resolution of ECM-associated proteins (i.e., macromolecules originally bound to the ECM but released upon the enzymatic digestion), matrikines (novel bioactive fragments which are liberated from the existing ECM components through the proteolysis), and matricryptins (novel bioactive fragments which are generated within the existing ECM components as soon as previously inactive, cryptic functional sites become exposed to microenvironment), which further shape the transitional phenotype of the immune cells inhabiting the BHVs. Synergistic action and heterogeneity of all mentioned factors entail distinct pathways of macrophage activation, multiple macrophage specifications, and a variety of secreted proinflammatory molecules which are partially trapped within the ECM. In addition, the engulfment of ECM fragments by macrophages leads to their transformation into multinucleated foreign-body giant cells and the further alteration of their surface markers. To summarise, the heterologous ECM and its digested snippets, broad repertoire of circulating macromolecules and electrolytes, and proinflammatory microenvironment determine the diversity of macrophage identities and the promiscuous expression of differentiation markers by host cells within the BHVs.

The most common example of the transitional phenotype in blood vessels and heart valves is the EndoMT, which is notable for the expression of both endothelial (e.g., CD31 or VEGFR2) and mesenchymal (e.g., α-SMA or fibroblast-associated protein) markers. The EndoMT has been reported as a spontaneous phenomenon in human umbilical vein endothelial cells cultured on an uncoated dish [40], as an inflammation- or ECM-induced molecular event in primary culture of VECs [41,42,43], as a physiological process indispensable for prenatal development of the heart [44,45,46], and also in the AVs obtained from the patients with CAS [47,48,49]. However, there have been no reports on whether the EndoMT occurs after the implantation of xenogeneic or tissue-engineered implants, including BHVs. In our study, VECs in the AVs occasionally showed the CD31^+^α-SMA^+^ phenotype indicative of the EndoMT, in line with previous results [47,48,50]. Whereas around ≈ 70% cells in BHVs were CD31^+^α-SMA^+^, they were not restricted to the valvular surface and were distributed over the valve. Although such phenotype does not indicate the EndoMT in this case (because the majority of cells within the BHVs are immune cells, but not ECs or mesenchymal cells), our electron microscopy approach (EM-BSEM, embedding of heavy-metal-stained whole-leaflet into epoxy resin with the following grinding, polishing, and backscattered scanning electron microscopy) confirmed the atypical appearance of endothelial-like cells covering the BHVs, confirming the previous results by our group [18] and contesting the concept on the postimplantation endothelialisation of BHVs [14,15,16,17]. Such endothelial-like cells also commonly exhibited the CD31^+^CD68^+^ phenotype and resembled macrophages rather than VECs, being rarely encountered in the AVs. 

Furthermore, immune cells within the BHVs did not have any clear phenotype from immunohistochemical or immunofluorescence staining, as most of them combined the expression of CD31 and CD45 or CD68 and α-SMA (albeit <30% cells were positive for α-SMA and CD45, and <15% cells displayed the simultaneous expression of CD31 and CD68). Most of these cells were colocalised with the degraded ECM and had a neutrophil or macrophage appearance from the EM-BSEM investigation. Moreover, a substantial proportion (30–40%) of macrophages had internalised ECM fragments in the cytoplasm, informative of their matrix-degrading ability and suggesting their proinflammatory activation. Presumably, these macrophages were initially CD45^+^ or CD68^+^ and then acquired the expression of nonspecific markers, such as the endothelial/platelet marker CD31 or the mesenchymal marker α-SMA, similar to tumour-associated macrophages [51] and alveolar macrophages of human lung allografts in patients with acute rejection [52] which gain CD31 expression, or to the macrophage-to-myofibroblast transition observed in kidney biopsies obtained from patients with renal fibrosis [53] or the recipients of kidney transplants [54]. In addition, sparse α-SMA^+^CD68^+^ cells were detected in the AVs as well, probably indicative of VICs which attained macrophage markers, or resident macrophages which underwent the macrophage-to-myofibroblast transition in a lipid-rich, proinflammatory, and procalcific milieu [55,56,57]. Earlier, it has been reported that circulating murine monocytes might express CD31 in conjunction with their canonical marker F4/80 and contribute to angiogenesis and wound healing [58]. Monocytes cultured for several days under angiogenic conditions or those overexpressing vascular endothelial growth factor (a potent proangiogenic molecule) lost CD14 and CD45, attaining CD31 expression instead [59,60,61], and the majority of CD14^+^ monocytes differentiated into ECs on biodegradable biomaterials [62]. However, monocytes cultured under serum deprivation or treated with transforming growth factor β1 differentiated into fibroblasts rather than ECs [63,64]. These results illustrate the high phenotypic plasticity of monocytes and macrophages, and point to the high probability of such a scenario in the implanted BHVs fabricated from glutaraldehyde- or ethylene-glycol-diglycidyl-ether-fixed bovine pericardium or porcine leaflets.

We propose that immunophenotyping cannot be considered as a valid approach to study cell populations within the BHVs, even in case of using sensitive and specific markers that are applied for studying the AVs (e.g., CD31, α-SMA, CD45, and CD68), as ≤20% cells in the BHVs were α-SMA^+^CD31^−^/CD45^−^/CD68^−^ or CD31^+^/α-SMA^−^/CD45^−^/CD68^−^. EM-BSEM can be advocated as an alternative option, as this electron microscopy technique allows the imaging of cells and the ECM with up to ×30,000 magnification, providing the high-quality visualisation of nuclei, the cell membrane, mitochondria, endoplasmic reticulum, Golgi apparatus, and engulfed fragments of the ECM [65]. In this study, interrogation of endothelial-like cells at the BHV surface by EM-BSEM disproved their endothelial phenotype in spite of the positive staining for CD31, which has been previously attributed to their endothelial identity [14,15,16,17]. Further, EM-BSEM furnishes an opportunity to distinguish quiescent and matrix-degrading macrophages by visualising digested, electron-dense ECM fragments inside the cytoplasm of the latter, whereas immunostaining for macrophage-polarisation markers, such as CD80, CD86, or inducible nitric oxide synthase (M1) or CD163 (M2), is capable of discerning proinflammatory (M1) and anti-inflammatory (M2) phenotypes, but is irrelevant for determining the matrix-degrading ability, which is crucial for the development of SVD. Noteworthy, multinucleated foreign-body giant cells and foam cells, which derive from actively phagocytosing macrophages, are also recognisable by EM-BSEM [18]. Another disadvantage of immunostaining in relation to BHVs is that their processing implies decalcification prior to sectioning on a microtome or cryostat to obtain high-quality sections. However, embedding of whole-tissue segments into epoxy resin during EM-BSEM keeps calcium deposits and their microenvironment for ultrastructural and elemental analysis. 

Therefore, the benefits of EM-BSEM for the analysis of BHVs include the ultrastructural imaging of cells and destructed ECM fibres, the visualisation of macrophage-mediated ECM digestion, and retained tissue integrity, whilst immunophenotyping fails to clearly define cell lineages amongst the host cells infiltrating the BHVs. In contrast to sectioning (a prerequisite of immunostaining), EM-BSEM provides an additional option of grinding the epoxy-resin-embedded tissue for any desirable thickness and the consecutive imaging of underlying layers with the subsequent three-dimensional (3D) reconstruction of the BHV. This can be used to investigate the BHV architecture, cell–cell and cell–ECM interactions, as well as the patterns of calcium-phosphate maturation and growth. Lifetime storage of epoxy-resin-embedded tissue can also be preferrable over fluorophores, which may fade over time if not using specific mountants.

Future perspectives in studying BHVs and corresponding applications of EM-BSEM include the development of an adhesive polymer sheath which is designed to protect the BHVs from circulating Ca^2+^ ions, proteases, immune cells, and bacteria entering the blood during transient bacteriemia. Such an envelope has already passed in vitro testing and is currently undergoing a preclinical trial in an ovine model to recapitulate human rheological conditions, and to simulate a “worst case model” in relation to prothrombotic conditions such as hypercoagulability (e.g., caused by genetic disorders). However, this trial is yet to be completed.

## 4. Materials and Methods

### 4.1. Sample Collection

We examined four BHVs tailored from bovine pericardium (NeoCor, Kemerovo, Russia), which were excised from the mitral position because of transprosthetic regurgitation, demanding redo surgery (i.e., BHV failure). Clinicopathological features of these patients have been described in our previous paper [18] and have been selected due to aggressive cell infiltration. The age of patients at the time of redo surgery was 70, 68, 67, and 60 years, and the duration of follow-up after the primary heart-valve replacement was 12, 14, 11, and 25 years, respectively. In addition, we investigated four AVs which have been removed because of calcific aortic valve disease (CAVD). The age of patients with CAVD was 66, 69, 72, and 66 years. Specimen collection was approved by the Local Ethical Committee of the Research Institute for Complex Issues of Cardiovascular Diseases (ethical approval code 15/2021, approved on 3 March 2021), and a written informed consent was provided by all study participants after receiving a full explanation of the study. The investigation was carried out in accordance with the Good Clinical Practice and a latest revision of the Declaration of Helsinki (2013).

Following the gross examination of the BHVs and AVs, we dissected deteriorated segments of the leaflets (i.e., those including mineral deposits, fibrotic masses, and/or tears), snap-froze them in an optimal-cutting-temperature medium (Tissue-Tek, 4583, Sakura Finetek, Tokyo, Japan), and cut them on a cryostat (6 µm sections, 6 sections per slide, Microm HM 525, Thermo Fisher Scientific, Waltham, MA, USA). Alternatively, deteriorated segments of the BHVs and AVs were fixed in 10% neutral phosphate-buffered formalin (B06-003, ErgoProduction, Saint Petersburg, Russia) and then used for ultrastructural analysis (see below).

### 4.2. Histological Examination and Immunostaining

General examination of the AVs and BHVs was performed by haematoxylin and eosin (H&E) staining, as described in [66]. For immunohistochemical staining, we fixed frozen sections in 4% paraformaldehyde (158127, Sigma-Aldrich, St. Louis, MO, USA) for 10 min, permeabilised in 0.1% Triton X-100 (T8787, Sigma-Aldrich, St. Louis, MO, USA) for 15 min, and then blocked endogenous peroxidase and nonspecific binding by the reagents from the Novolink Polymer Detection System (RE7150-CE, Leica Biosystems, Wetzlar, Germany) according to the manufacturers’ protocol. Staining was conducted using the primary, antigen-specific antibodies to CD31 (1:100 dilution, MAB1393-I (mouse antihuman, monoclonal), Sigma-Aldrich, St. Louis, MO, USA, or 1:200 dilution, ab182981 (rabbit antihuman, monoclonal), Abcam, Cambridge, UK), VEGFR2 (1:200 dilution, ab39256 (rabbit antihuman, polyclonal), Abcam, Cambridge, UK), α-SMA (1:1000 dilution, ab7817 (mouse antihuman, monoclonal), Abcam, Cambridge, UK; 1:100 dilution, ab5694 (rabbit antihuman, polyclonal), Abcam, Cambridge, UK), VIM (1:1000 dilution, ab16700 (rabbit antihuman, monoclonal), Abcam, Cambridge, UK), CD45 (1:1500 dilution, ab10558 (rabbit antihuman, polyclonal), Abcam, Cambridge, UK), and CD68 (1:200 dilution, ab955 (mouse antihuman, monoclonal), Abcam, Cambridge, UK), and secondary species-specific antibodies to mouse or rabbit from the Novolink Polymer Detection System (RE7150-CE, Leica Biosystems, Wetzlar, Germany) according to the manufacturers’ protocol. Modified Mayer’s haematoxylin from this kit was used as a counterstain. Coverslips were mounted with Vitrogel (HM-VI-A500, ErgoProduction, St. Petersburg, Russia). Visualisation was carried out by light microscopy (AxioImager.A1 microscope, EC Plan-Neofluar 20×/0.50 or EC Plan-Neofluar 40×/0.75 M27 objectives and AxioVision software Rel. 4.8, Carl Zeiss, Oberkochen, Germany). Intact (nonimplanted) bovine pericardium treated similarly to the excised AVs and BHVs was employed as a negative control. Each slide had one section without either primary (antigen-specific) or secondary (species-specific) antibodies to be used as a technical control.

For immunofluorescence staining, we fixed and permeabilised frozen sections as described above, but the blocking of the nonspecific binding was made with 1% bovine serum albumin (P091E, PanEco, Moscow, Russia) for 1 h. Incubation with primary, antigen-specific antibodies was performed as above (16 h at +4 °C) whilst in incubation with secondary, species-specific donkey antimouse Alexa Fluor 488- and donkey antirabbit Alexa Fluor 555-labeled antibodies (1:500 dilution, ab150105 and ab150074). We employed the following antibody combinations (Table 1):

(1) Mouse antihuman CD31 (1:100 dilution, MAB1393-I, Sigma-Aldrich, St. Louis, MO, USA) and rabbit antihuman VEGFR2 (1:200 dilution, ab39256, Abcam, Cambridge, UK);

(2) Mouse antihuman α-SMA (1:1000 dilution, ab7817, Abcam, Cambridge, UK) and rabbit antihuman VIM (1:1000 dilution, ab16700, Abcam, Cambridge, UK);

(3) Mouse antihuman CD68 (1:200 dilution, ab955, Abcam, Cambridge, UK) and rabbit antihuman CD45 (1:1500 dilution, ab10558, Abcam, Cambridge, UK);

(4) Mouse antihuman CD31 (1:100 dilution, MAB1393-I, Sigma-Aldrich, St. Louis, MO, USA) and rabbit antihuman α-SMA (1:100 dilution, ab5694, Abcam, Cambridge, UK);

(5) Mouse antihuman CD31 (1:100 dilution, MAB1393-I, Sigma-Aldrich, St. Louis, MO, USA) and rabbit antihuman CD45 (1:1500 dilution, ab10558, Abcam, Cambridge, UK);

(6) Mouse antihuman CD68 (1:200 dilution, ab955, Abcam, Cambridge, UK) and rabbit antihuman CD31 (1:200 dilution, ab182981, Abcam, Cambridge, UK);

(7) Mouse antihuman α-SMA (1:1000 dilution, ab7817, Abcam, Cambridge, UK) and rabbit antihuman CD45 (1:1500 dilution, ab10558, Abcam, Cambridge, UK);

(8) Mouse antihuman CD68 (1:200 dilution, ab955, Abcam, Cambridge, UK) and rabbit antihuman α-SMA (1:100 dilution, ab5694, Abcam, Cambridge, UK).

After the staining, sections were treated with an autofluorescence quencher (2160, Merck Millipore, Burlington, MA, USA) according to the manufacturer’s protocol and then incubated with nuclear counterstain DAPI (0.1 μg/mL, D9542, Sigma-Aldrich, St. Louis, MO, USA) for 30 min. Slides were mounted with the ProLong Gold Antifade Mountant (P36930, Thermo Fisher Scientific, Waltham, MA, USA) and examined by confocal laser scanning microscopy (LSM 700 microscope, EC Plan-Neofluar 20x/0.50 M27 and Plan-Apochromat 40×/1.3 Oil DIC M27 objectives and ZEN 2012 SP1 8.1 software, Carl Zeiss, Oberkochen, Germany).

### 4.3. Electron Microscopy

For the ultrastructural analysis, we applied the EM-BSEM approach, which was described in detail [65]. Briefly, deteriorated segments of BHVs and AVs were fixed in 10% neutral phosphate-buffered formalin (B06-003, ErgoProduction, Saint Petersburg, Russia), postfixed and stained in 1% phosphate-buffered osmium tetroxide (OsO_4_, 19110, Electron Microscopy Sciences, Hatfield, PA, USA), dehydrated in an ascending ethanol series (Kemerovo Pharmaceutical Plant, Kemerovo, Russia) and isopropanol (06-002, ErgoProduction, Saint Petersburg, Russia), stained in 2% alcoholic uranyl acetate (22400-2, Electron Microscopy Sciences, Hatfield, PA, USA), impregnated with acetone (6-09-20-03-83, EKOS-1, Moscow, Russia): epoxy resin (1:1) and epoxy resin (Araldite 502, 13900, Electron Microscopy Sciences, Hatfield, PA, USA), embedded into epoxy resin, grinded, polished, and counterstained with Reynolds’s lead citrate (17810, Electron Microscopy Sciences, Hatfield, PA, USA), sputter-coated with carbon (Leica EM ACE200, Leica Microsystems, Wetzlar, Germany), and visualised by means of backscattered scanning electron microscopy (S-3400N, Hitachi, Tokyo, Japan) in accordance with an EM-BSEM procedure similar to the previously published literature [11,18,66,67,68,69].

### 4.4. Quantitative Image Analysis

Quantitative image analysis of confocal microscopy images from the AVs and BHVs was carried out by counting single-positive (i.e., those stained with either of fluorescent-labelled antibodies) and double-positive (i.e., those stained by both fluorescent-labelled antibodies) cells and calculating their proportions from all positively stained cells in 10 fields of view for each of the following antibody combinations: 

(1) Mouse antihuman CD31 (1:100 dilution, MAB1393-I, Sigma-Aldrich, St. Louis, MO, USA) and rabbit antihuman α-SMA (1:100 dilution, ab5694, Abcam, Cambridge, UK); 

(2) Mouse antihuman CD31 (1:100 dilution, MAB1393-I, Sigma-Aldrich, St. Louis, MO, USA) and rabbit antihuman CD45 (1:1500 dilution, ab10558, Abcam, Cambridge, UK); 

(3) Mouse antihuman CD68 (1:200 dilution, ab955, Abcam, Cambridge, UK) and rabbit antihuman CD31 (1:200 dilution, ab182981, Abcam, Cambridge, UK); 

(4) Mouse antihuman α-SMA (1:1000 dilution, ab7817, Abcam, Cambridge, UK) and rabbit antihuman CD45 (1:1500 dilution, ab10558, Abcam, Cambridge, UK); 

(5) Mouse antihuman CD68 (1:200 dilution, ab955, Abcam, Cambridge, UK) and rabbit antihuman α-SMA (1:100 dilution, ab5694, Abcam, Cambridge, UK).

In addition, we calculated the proportions of CD31^+^/CD45^+^ and α-SMA^+^/CD45^+^ to all CD45^+^ cells, and the proportions of CD68^+^/CD31^+^ and CD68^+^/α-SMA^+^ to all CD68^+^ cells. As we computed median proportions, the total sum of the proportions was not equal to 100%.

The rationale behind choosing 10 fields of view was based on: (1) the heterogeneous distribution of cells across the BHVs, as their highest density was observed within the degraded ECM, whereas intact areas were devoid of host cells; (2) multiple crosslinks formed in bovine pericardium by glutaraldehyde or ethylene glycol diglycidyl ether, which led to significant autofluorescence of the biomaterial and a high background even at confocal microscopy. We selected the images with the lowest background and dense foci of cellular infiltration (*n* = 2–3 from each AV or BHV). For the count of positively stained cells, we selected the fields of view containing the degenerative ECM around the calcifications (AVs and BHVs), within the fibrotic-thickening areas (AVs), and near the perforations (BHVs), as only these fields of view contained all studied cell populations.

### 4.5. Statistical Analysis

Statistical analysis was performed using GraphPad Prism 8 (GraphPad Software, San Diego, CA, USA). For descriptive statistics, data were represented by the proportions, the median, and 25th and 75th percentiles. Groups were compared by the Mann–Whitney U-test. The *p*-values ≤ 0.05 were regarded as statistically significant.

## 5. Conclusions

Although immunostaining is the technique of choice for the molecular mapping of well-defined cell populations composing the AVs (such as CD31^+^VEGFR2^+^ VECs or α-SMA^+^VIM^+^ VICs), it is not informative for studying BHVs which are inhabited by transitional cell populations, which typically combine the expression for ≥2 markers of distinct lineages (e.g., CD31 together with CD45, or CD31, CD45, or CD68, along with α-SMA). For the comprehensive examination of BHVs, we suggest EM-BSEM (EMbedding and Backscattered Scanning Electron Microscopy), an approach integrating the assessment of the macrophage-matrix-degrading activity, the high-definition imaging of cell membranes, organelles, and the ECM, the nondestructive analysis of calcified tissues, the layer-by-layer 3D reconstruction of embedded tissue, and the lifetime storage of the samples.

## Figures and Tables

**Figure 1 ijms-24-13602-f001:**
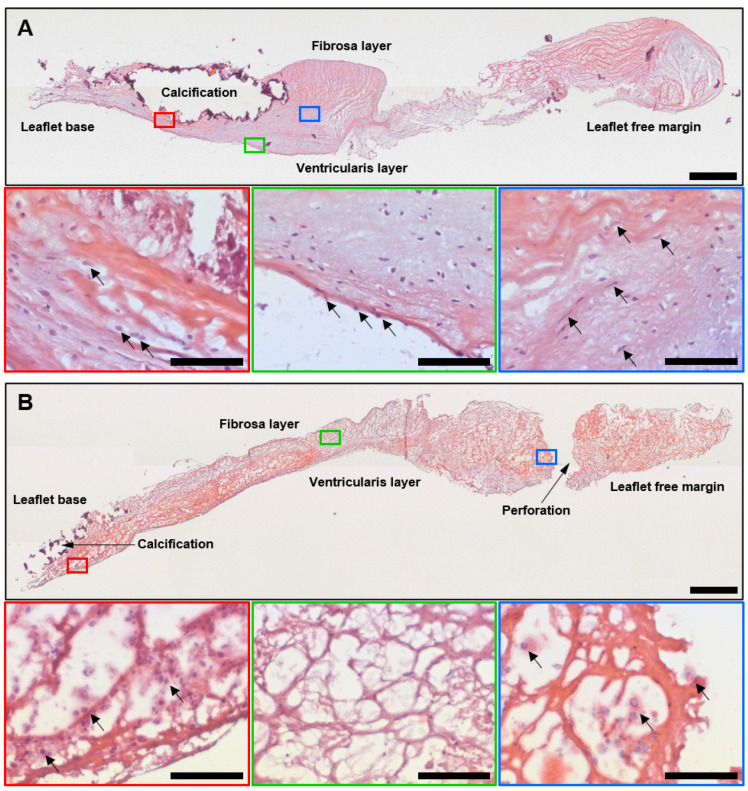
Histological examination of calcified AVs and SVD-affected BHVs. (**A**) AVs excised from the patients with CAS because of critical leaflet thickening and severe calcification. Aside from calcification, the fibrotic ECM retained its integrity. Macrophage spread was limited to areas with a degraded ECM around the calcium deposits (demarcated by a red contour; macrophages are indicated by black arrows), whilst the AV surface (demarcated by a green contour) was covered by VECs (indicated by black arrows) and the fibrotic ECM (demarcated by a blue contour) was populated by VICs (indicated by black arrows). The fibrosa layer (aortic, or outflow side) is on the top, whereas the ventricularis layer (ventricular, or inflow side) is at the bottom of the section; (**B**) In comparison with AVs, BHVs suffered from ECM disintegration and the fibres were loosened. Immune cells were located at the base of the leaflets and near the mineral deposits (demarcated by a red contour; immune cells are indicated by black arrows). Areas devoid of host cells (indicated by a green contour) alternated with cellular infiltrations associated with perforations and tears (indicated by a blue contour; immune cells are indicated by black arrows). The fibrosa layer (aortic, or outflow side) is on the top, whereas the ventricularis layer (ventricular, or inflow side) is at the bottom of the section. The bottom images within each group (demarcated by red, green, and blue contours; ×200 magnification, scale bar: 100 μm) are close-ups of the top images (overview; scale bar: 1000 μm). Haematoxylin and eosin staining.

**Figure 2 ijms-24-13602-f002:**
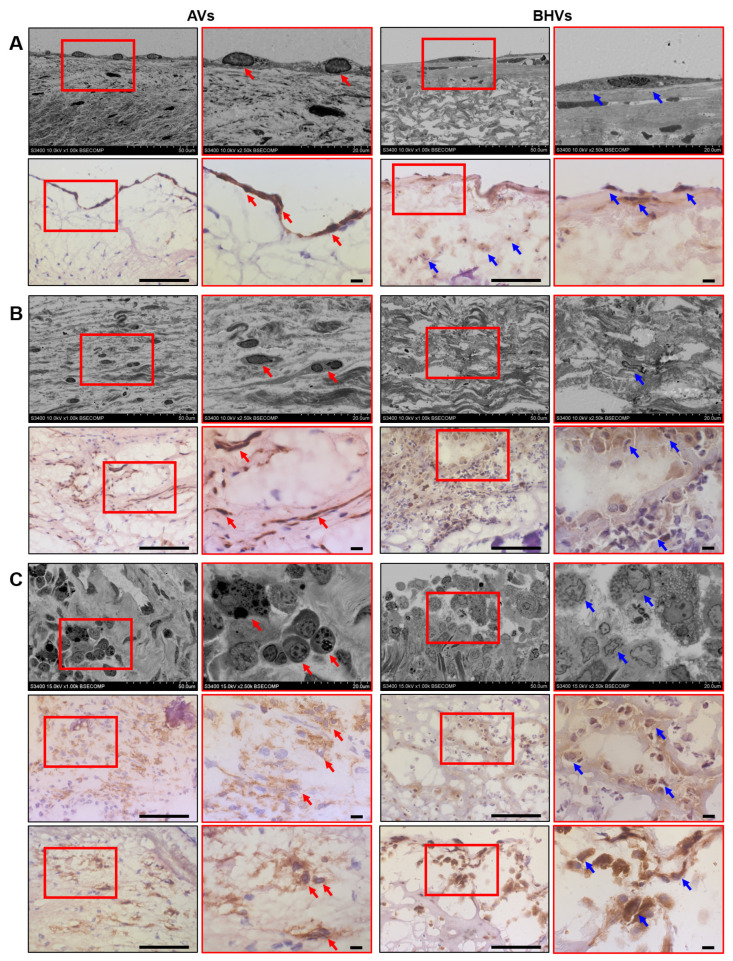
Side-by-side comparison of ultrastructural (EM-BSEM) and immunohistochemical analysis of calcified AVs and SVD-affected BHVs. (**A**) ECs had distinct appearance in AVs and BHVs: AVs (left images) were covered with a monolayer of VECs (indicated by red arrows) that had an elongated shape and a large nucleus oriented towards the flow, and were positive for the canonical EC marker CD31, whilst BHVs (right images) were lined with endothelial-like cells (indicated by blue arrows) that had a bigger size, lower nucleus-to-cytoplasm ratio, and contained noticeable inclusions, being also slightly positive for CD31. Note the concordance of ultrastructural and immunohistochemical analysis in the AVs and the detailed rather than vague phenotype of endothelial-like cells in the BHVs when examined by EM-BSEM, but not immunohistochemical staining. EM-BSEM: top images, immunohistochemical staining: bottom images; (**B**) Mesenchymal cells were abundant in the AVs (VICs, left images, indicated by red arrows) and were positive for α-SMA, but were rarely encountered in the BHVs, where they showed lower α-SMA expression (myofibroblasts, right images, indicated by blue arrows). EM-BSEM: top images, immunohistochemical staining: bottom images; (**C**) Although macrophages were observed in both AVs and BHVs, and were detected mostly around calcium deposits and in the degraded ECM, the AVs (left images) were infiltrated exclusively by canonical, CD45^+^, and CD68^+^ macrophages (indicated by red arrows), which significantly differed from the large ECM-digesting macrophages and multinucleated foreign-body giant cells (indicated by blue arrows) within the BHVs, which were also positive for CD45 and CD68 (right images). Note that the ultrastructural analysis of the macrophages permits the assessment of their ECM-digesting activity, whilst immunohistochemistry does not. Note the fibrotic ECM of the AVs and the loosened ECM of the BHVs, which is clearly visualised in the EM-BSEM but not the immunohistochemical staining. EM-BSEM: top images, immunohistochemical staining: bottom images. The right images within each group (demarcated by a red contour; EM-BSEM: ×2500 magnification, scale bar: 20 μm; immunohistochemistry: ×1000 magnification, scale bar: 10 μm) are close-ups of the corresponding left images (overview; EM-BSEM: ×1000 magnification, scale bar: 50 μm; immunohistochemistry: ×400 magnification, scale bar: 100 μm).

**Figure 3 ijms-24-13602-f003:**
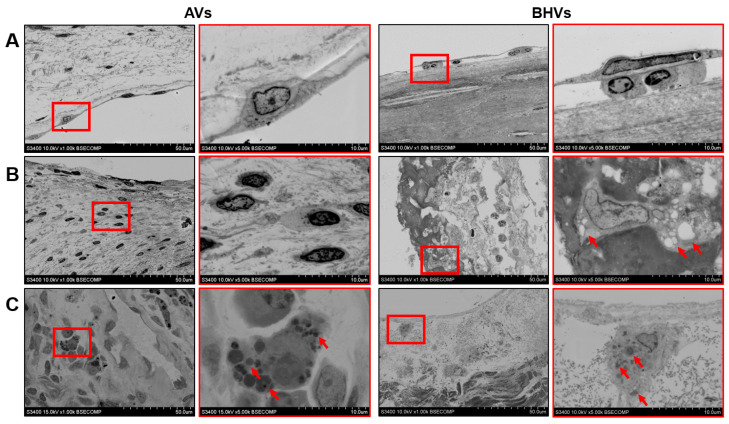
High-magnification ultrastructural analysis of calcified AVs and degenerated BHVs. (**A**) VECs at the surface of the AVs (left images) had an elongated shape and formed a continuous monolayer by contacting through the tight junctions, whilst endothelial-like cells lining the BHVs (right images) were associated with adhering monocytes and their layer was intermittent; (**B**) VICs within the AVs (left images) generally had a high nucleus-to-cytoplasm ratio and were devoid of any inclusions, whereas the sparse fibroblasts or myofibroblasts frequently had lipid droplets (indicated by red arrows) and digested ECM fragments; (**C**) Digesting activity of the macrophages was indicated by internalised and partitioned ECM fragments (indicated by red arrows) and by the lysed or loosened ECM surrounding such macrophages. Such phenomenon was infrequent in the AVs (left images) but common for the BHVs (right images). The right images within each group (demarcated by red contour, ×5000 magnification, scale bar: 10 μm) are close-ups of the corresponding left images (overview, ×1000 magnification, scale bar: 50 μm).

**Figure 4 ijms-24-13602-f004:**
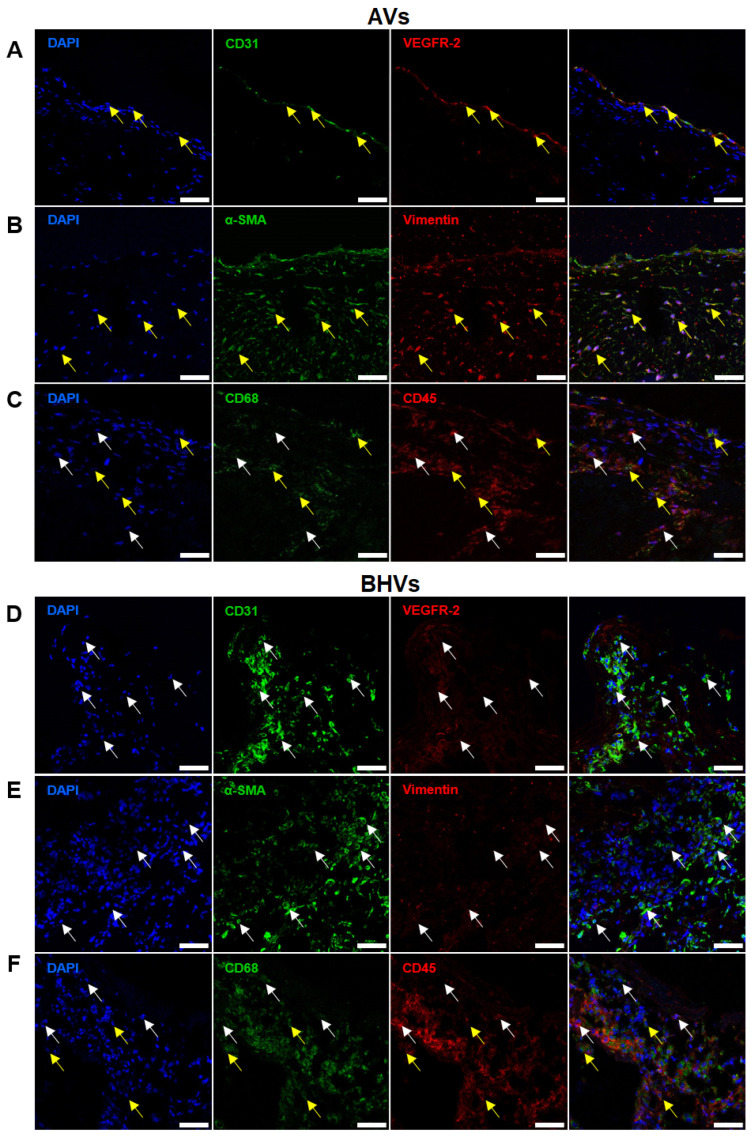
Immunofluorescence staining of AVs and BHVs with antibodies to EC (CD31 and VEGFR2), contractile (α-SMA and VIM), and immune-cell (CD45 and CD68) markers. (**A**) AVs, staining for CD31 (green) and VEGFR2 (red), and ECs (CD31^+^VEGFR2^+^ cells) are indicated by yellow arrows; (**B**) AVs, staining for α-SMA (green) and VIM (red), and VICs (α-SMA^+^VIM^+^ cells) are indicated by yellow arrows; (**C**) AVs, staining for CD68 (green) and CD45 (red), and CD45^+^CD68^-^ cells are indicated by white arrows, and CD45^+^CD68^+^ cells are indicated by yellow arrows; (**D**) BHVs, staining for CD31 (green) and VEGFR2 (red), and CD31^+^VEGFR2^-^ cells are indicated by white arrows; (**E**) BHVs, staining for α-SMA (green) and VIM (red), and α-SMA^+^VIM^-^ cells indicated by white arrows; (**F**) BHVs, staining for CD68 (green) and CD45 (red), and CD45^+^CD68^−^ and CD68^+^CD45^−^ cells are indicated by white arrows, and CD45^+^CD68^+^ cells are indicated by yellow arrows. Nuclei are counterstained with 4′,6-diamidino-2-phenylindole (DAPI, blue). Magnification: ×400, scale bar: 50 μm.

**Figure 5 ijms-24-13602-f005:**
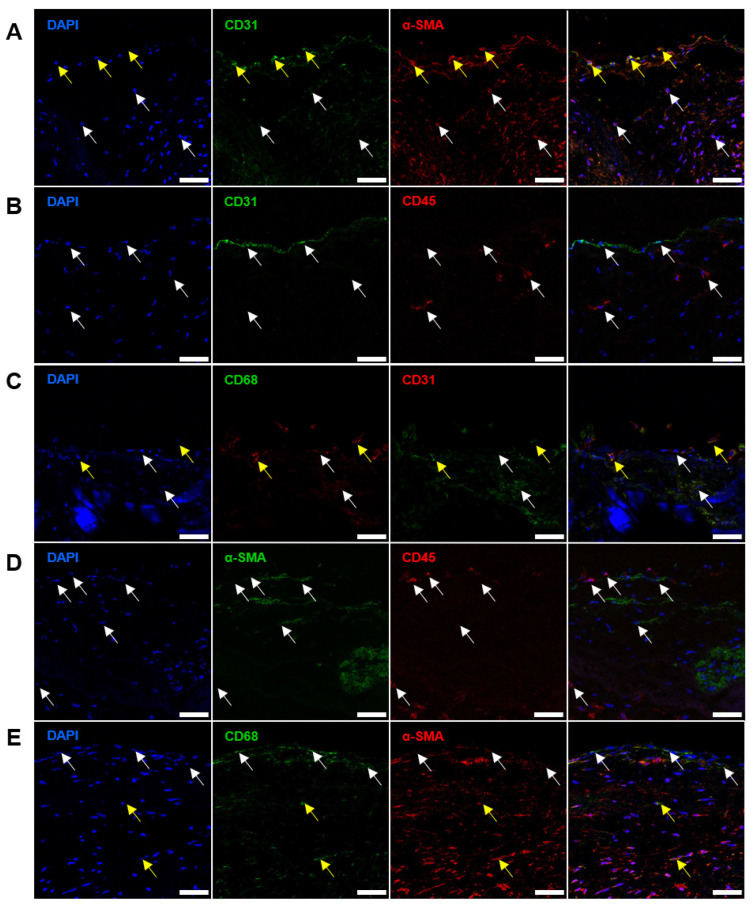
Immunofluorescence staining of AVs with antibodies to: (**A**) CD31 (green) and α-SMA (red), α-SMA^+^CD31^−^ cells are indicated by white arrows, and CD31^+^α-SMA^+^ are indicated by yellow arrows; (**B**) CD31 (green) and CD45 (red), and CD31^+^CD45^−^ and CD45^+^CD31^−^ cells are indicated by white arrows; (**C**) CD68 (green) and CD31 (red), CD31^+^CD68^−^ cells are indicated by white arrows, and CD31^+^CD68^+^ cells are indicated by yellow arrows; (**D**) α-SMA (green) and CD45 (red), and α-SMA^+^CD45^−^ and CD45^+^α-SMA^−^ are indicated by white arrows; (**E**) CD68 (green) and α-SMA (red), CD68^+^α-SMA^−^ cells are indicated by white arrows, and CD68^+^α-SMA^+^ cells are indicated by yellow arrows. Nuclei are counterstained with 4′,6-diamidino-2-phenylindole (DAPI) (blue). Magnification: ×400, scale bar: 50 μm.

**Figure 6 ijms-24-13602-f006:**
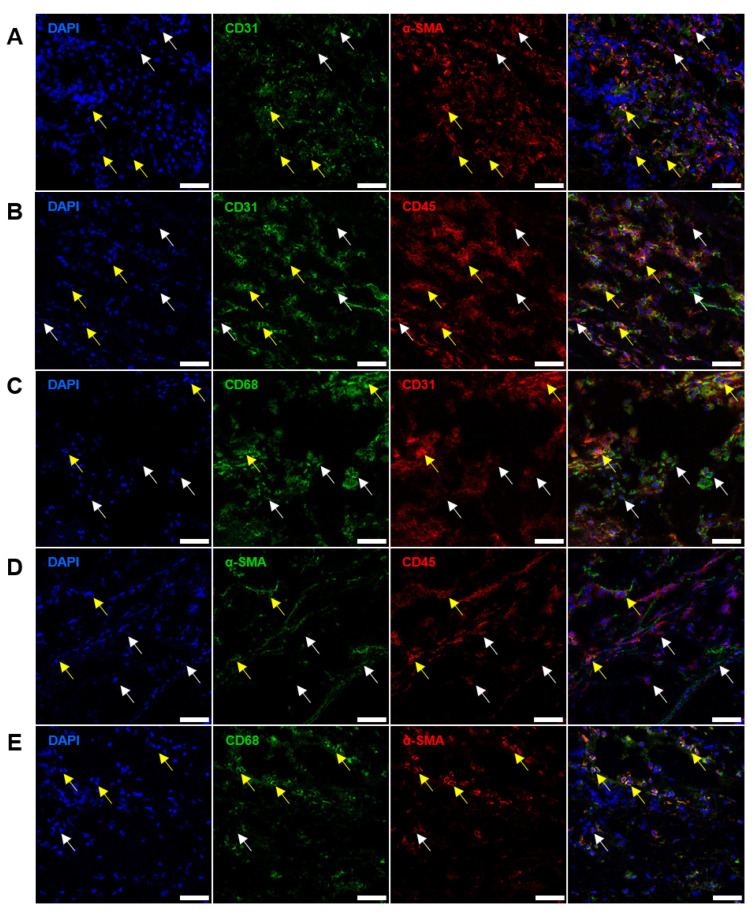
Immunofluorescence staining of BHVs with antibodies to: (**A**) CD31 (green) and α-SMA (red), α-SMA^+^CD31^−^ cells are indicated by white arrows, and CD31^+^α-SMA^+^ are indicated by yellow arrows; (**B**) CD31 (green) and CD45 (red), CD31^+^CD45^−^ and CD45^+^CD31^−^ cells are indicated by white arrows, and CD31^+^CD45^+^ cells are indicated by yellow arrows; (**C**) CD68 (green) and CD31 (red), CD68^+^CD31^−^ cells are indicated by white arrows, and CD68^+^CD31^+^ cells are indicated by yellow arrows; (**D**) α-SMA (green) and CD45 (red), α-SMA^+^CD45^−^ and CD45^+^α-SMA^−^ cells are indicated by white arrows, and α-SMA^+^CD45^+^ cells are indicated by yellow arrows; (**E**) CD68 (green) and α-SMA (red), α-SMA^+^CD68^−^ cells are indicated by white arrows, and CD68^+^α-SMA^+^ cells are indicated by yellow arrows. Nuclei are counterstained with DAPI (blue). Magnification: ×400, scale bar: 50 μm.

**Figure 7 ijms-24-13602-f007:**
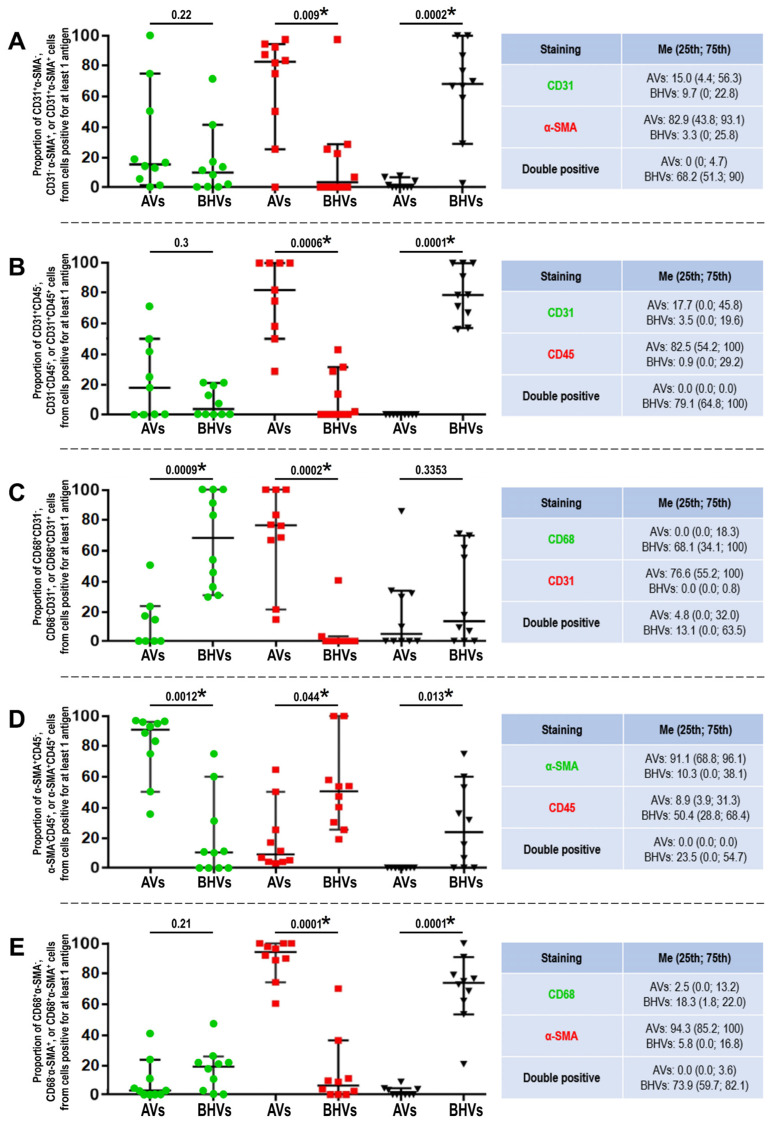
Quantitative image analysis (count of positive cells) of immunofluorescence stainings performed in the AVs and BHVs. (**A**) CD31 and α-SMA staining, the proportions of CD31^+^α-SMA^−^, CD31^−^α-SMA^+^, and CD31^+^α-SMA^+^ cells on the images are represented by green dots, red squares, and black inverted triangles, respectively; (**B**) CD31 and CD45 staining, the proportions of CD31^+^CD45^−^, CD31^−^CD45^+^, and CD31^+^CD45^+^ cells on the images are represented by green dots, red squares, and black inverted triangles, respectively; (**C**) CD68 and CD31 staining, the proportions of CD68^+^CD31^−^, CD68^−^CD31^+^, and CD68^+^CD31^+^ cells on the images are represented by green dots, red squares, and black inverted triangles, respectively; (**D**) α-SMA and CD45 staining, the proportions of α-SMA^+^CD45^−^, α-SMA^−^CD45^+^, and α-SMA^+^CD45^+^ cells on the images are represented by green dots, red squares, and black inverted triangles, respectively; (**E**) CD68 and α-SMA staining, the proportions of CD68^+^α-SMA^−^, CD68^−^α-SMA^+^, and CD68^+^α-SMA^+^ cells on the images are represented by green dots, red squares, and black inverted triangles, respectively; Colours of the dots on each plot (green, red, and black) correspond to the antigen indicated in the table at the right side. Each dot, square, or inverted triangle on the plots represents one image (*n* = 9–10 measurements per group). Whiskers of the 25th–75th percentiles and centre lines indicate the median. The *p* values are provided above the plots, and statistically significant *p* values are marked by asterisks; Mann–Whitney U-test.

**Figure 8 ijms-24-13602-f008:**
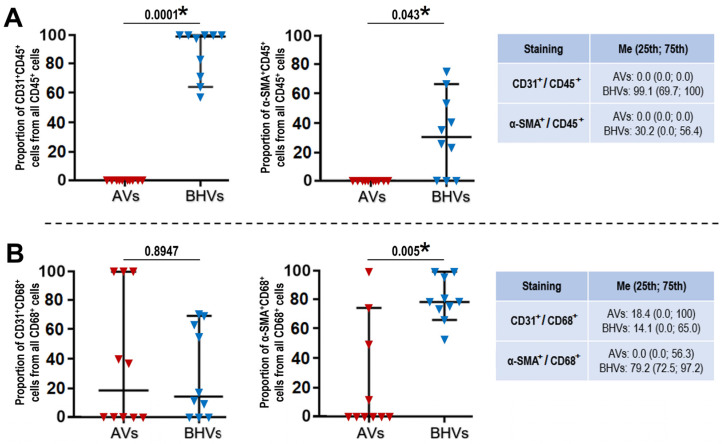
Quantitative image analysis (calculation of the proportions of CD31^+^/CD45^+^, α-SMA^+^/CD45^+^, CD68^+^/CD31^+^, and CD68^+^/α-SMA^+^ to all CD45^+^ and CD68^+^ cells) of immunofluorescence stainings performed in the AVs and BHVs. (**A**) CD31/CD45 and α-SMA/CD45 stainings showing the proportions of CD31^+^CD45^+^ cells (left image) and α-SMA^+^CD45^+^ cells (right image) in AVs (red inverted triangles) and BHVs (blue inverted triangles) to all CD45^+^ cells; (**B**) CD31/CD68 and α-SMA/CD68 stainings showing the proportions of CD31^+^CD68^+^ cells (left image) and α-SMA^+^CD68^+^ cells (right image) in AVs (red inverted triangles) and BHVs (blue inverted triangles) to all CD68^+^ cells. Each inverted triangle on the plots represents one image (*n* = 9–10 measurements per group). Whiskers of the 25th–75th percentiles and centre lines indicate the median. The *p* values are provided above the plots, and statistically significant *p* values are marked by asterisks; Mann–Whitney U-test.

**Table 1 ijms-24-13602-t001:** Antibody combinations used for the immunofluorescence staining.

Mouse Primary Antibody (Labelled by Donkey Antimouse Alexa Fluor 488-Labeled Secondary)			Rabbit Primary Antibody (Labelled by Donkey Antimouse Alexa Fluor 555-Labeled Secondary)		
Catalogue Number and Manufacturer	Dilution	Antigen	Catalogue Number and Manufacturer	Dilution	Antigen
MAB1393-I, Sigma-Aldrich	1:100	CD31	ab39256, Abcam	1:200	VEGFR2
ab7817, Abcam	1:1000	α-SMA	ab16700, Abcam	1:1000	VIM
ab955, Abcam	1:200	CD68	ab10558, Abcam	1:1500	CD45
MAB1393-I, Sigma-Aldrich	1:100	CD31	ab5694, Abcam	1:100	α-SMA
MAB1393-I, Sigma-Aldrich	1:100	CD31	ab10558, Abcam	1:1500	CD45
ab955, Abcam	1:200	CD68	ab182981, Abcam	1:200	CD31
ab7817, Abcam	1:1000	α-SMA	ab10558, Abcam	1:1500	CD45
ab955, Abcam	1:200	CD68	ab5694, Abcam	1:100	α-SMA

## Data Availability

The datasets used and analysed during the current study are available from the corresponding author upon reasonable request.

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
