# Peer review of "Embedding and Backscattered Scanning Electron Microscopy (EM-BSEM) Is Preferential over Immunophenotyping in Relation to Bioprosthetic Heart Valves"

_ijms, 2023, doi:10.3390/ijms241713602_

Round 1

Reviewer 1 Report

The current manuscript titled in “Embedding and backscattered scanning electron microscopy (EM-BSEM) is preferential over immunophenotyping in relation to bioprosthetic heart valves”, The authors confirmed the transitional phenotype of most immune cells within the BHVs, and that EM-BSEM could be an appropriate and superior technique to study the causes of BHV failure. Overall, minor revisions should be done before publication. The following issues still exist:

1.     The manuscript is interesting and completed, but the images can be improved to merge or combine some ones in order to enrich and refine the manuscript.

2.     It is suggested that subheadings be added to each paragraph in the results. I am not sure about the format of the journal. Please double check it.

3.     Where there are significant differences in the statistical graphs, please mark them with a significance symbol in order to facilitate readers.

4.     Discussion can be improved to provide more details based on the results and perspectives in the future plan. Any recent references can be added?

5.     Some typo errors should be paid more attention, proper proofreading should be completed prior to publication.

The language use can be acceptable, however minor revision should be done for some typo errors.

Author Response

We sincerely thank the reviewer for the constructive criticism and valuable notes, which collectively helped us to improve the paper. Please see the attachment.

Reviewer 2 Report

The aim of this study is to introduce a new approach to conduct ultrastructural analysis of organelles and ECM in bioprosthetic heart valves using a non-destructive electron microscopy technique on whole specimens. The authors claim that this EM-BSEM approach has advantages over immunophenotyping, particularly in distinguishing between quiescent and matrix-degrading macrophages, foam cells, and multinucleated giant cells, while preserving tissue integrity.

The study is structured clearly, with the team choosing two types of samples (Calcified-AV and SVD-BHV), characterizing them using traditional methods such as immunohistochemistry and immunofluorescence staining, and comparing them with the EM-BSEM characterization. Their data, results, and discussion sections are comprehensive and detailed, providing valid evidence for their claim that uncertain immunophenotype was rarely encountered in the AVs, while it was commonly found in BHVs.

However, one major concern raised by the reviewer is that the manuscript does not provide enough evidence to support the claim that EM-BSEM is preferential over immunophenotyping in relation to bioprosthetic heart valves. The only figure about EM-BSEM (Figure 2) seems to suggest that it is an optional tool to verify or validate the hypothesis/assumption set up from traditional immunohistology/immunofluorescence characterization. Therefore, the reviewer suggests that the authors should provide more input/figures about EM-BSEM and compare it side-by-side with immunophenotyping to highlight the findings that immunophenotyping cannot demonstrate. Alternatively, the authors could consider editing their title to make it more humble and less ambitious.

In addition to this major concern, the reviewer also lists other questions that arise from reading the paper. The authors are encouraged to address these questions properly in the revised work. Good luck!

·       Line 74-76: Can you explain the mechanism of EM-BSEM and its advantages compared to immunophenotyping? Does it only detect macrophages or other types of cells as well?

·       Regarding the AV surface, whether it is calcified aortic leaflets or BHV, it should be referred to as the ventricularis layer on the in-flow side, right? If so, Please note this at the beginning, especially considering that Fig 1A and 1B have the Spongiosa side on the top.

·       In line 148-151, the distribution of immune cells in the BHV (Fig 3, CD-45 and CD-68) seems quite similar to the ones in Figure 2C-3 and Figure 2C-4, with the distribution of macrophages. Could you specify which two groups of images you are about to compare? To the reviewer, your Figure 2A can be compared to Fig3-CD31 and Figure 2C can be compared to Fig 3-CD68. However, due to the amplification, it’s hard to tell the morphology of the cells in your immunohistochemical staining images. Please label the sub-panel figures as A, B, C...

·       In line 244-246, where did you count the total cells? Did you choose the images from different sites of the leaflets, measure them, and count them globally, or did you choose a specific area, like the calcium-accumulated area, to count the cells locally?

·       In line 246, where did you count CD68/CD31, CD68/alpha-SMA, not CD31/CD68, alpha-SMA/CD68, like what you did to CD45? The latter one will make more sense if you claimed that "in relation to the total count of CD45 and CD68."

·       In line 272, the reason for BHV structural valve degeneration is not easy to conclude in one sentence. However, considering most of the BHVs on the market have to go through the treatment on the bovine pericardium, they are intrinsically susceptible to calcification due to the residues from Glutaraldehyde and cellular components, which impact the cell expressing and lead the calcium deposition. What's more, non-calcific mechanism could also be the reason to lead SVD albeit it's beyond this manuscript's scope, FYI. In terms of this discussion, Xue et al (doi: 10.1093/cvr/cvac002) provides a valuable explanation to this phenomenon and could be helpful to expand your discussion about this topic.

·       In line 377, EM-BSEM could be a powerful tool to show higher magnification, high-quality visualization, etc. in cited reference but your work didn't demonstrate those points. Your Figure 2 is more like to verify/validate what you found in immunohistology and immunofluorescence staining. Although it displayed some extra information, it seems not as powerful as you introduced in this discussion session.

Author Response

(The authors gave the same response as above.)
